# Maintenance of pure hybridogenetic water frog populations: Genotypic variability in progeny of diploid and triploid parents

Dmitrij Dedukh[1], Sergey Riumin[1,2], Krzysztof Kolenda[3], Magdalena Chmielewska[3], Beata Rozenblut-Kościsty[3], Mikołaj Kaźmierczak[3], Maria Ogielska[3], Alla Krasikova[1]*

1 Department of Cytology and Histology, Saint-Petersburg State University, Saint-Petersburg, Russia,
2 Raisa Gorbacheva Memorial Research Institute for Pediatric Oncology, Hematology and Transplantation, Pavlov First St. Petersburg State Medical University, Saint-Petersburg, Russia, 3 Amphibian Biology Group, Department of Evolutionary Biology and Conservation of Vertebrates, Faculty of Biological Sciences, University of Wrocław, Wrocław, Poland

* a.krasikova@spbu.ru

**Data Availability Statement:** All relevant data are within the article and its Supporting Information files.

## Abstract

An intriguing outcome of hybridisation is the emergence of clonally and hemiclonally reproducing hybrids, that can sustain, reproduce, and lead to the emergence of polyploid forms. However, the maintenance of diploid and polyploid hybrid complexes in natural populations remains unresolved. We selected water frogs from the *Pelophylax esculentus* complex to study how diploid and triploid hybrids, which reproduce hemiclonally via hybridogenesis, are maintained in natural populations. During gametogenesis in diploid hybrids, one of the parental genomes is eliminated, and the remaining genome is endoreplicated. In triploid hybrids, the single-copy genome is typically eliminated, while genome endoreplication does not occur. To investigate how diploid and triploid hybrid frogs reproduce in populations without parental species, we crossed these hybrid animals from two separate pure hybrid populations located in Poland. Using cytogenetic analysis of tadpoles that emerged from the crosses, we established which gametes were produced by parental hybrids. The majority of hybrid females and hybrid males produced one type of gamete with the *P. ridibundus* genome. However, in both studied populations, approximately half of the diploid and triploid hybrids simultaneously produced gametes with different genome compositions and ploidy levels, specifically, the *P. ridibundus* and *P. lessonae* genomes, as well as diploid gametes with genomes of both parental species. Triploid hybrid males and females mostly produced haploid gametes with the *P. lessonae* genome; however, gametes with the *P. ridibundus* genome have also been observed. These results suggest that not all hybrids follow the classical hybridogenetic reproduction program and reveal a significant level of alterations in the gametogenesis pathways. In addition, we found a variable survival rate of particular progeny genotypes when we crossed hybrid females with different males suggesting the important role of postzygotic barriers on the maintenance of pure hybrid systems. We suggest that the observed variability in produced gametes and the different survival rate of the progeny with certain genotypes is crucial for the existence of pure hybrid systems.

**Funding:** D.D. Russian Science Foundation (https://rscf.ru/en/) № 20-74-00030 M.O. Polish National Science Centre (https://www.ncn.gov.pl/en) no.NCN 2012/07/B/NZ3/02563 The funders had no role in study design, data collection and analysis, decision to publish, or preparation of the manuscript.

**Competing interests:** The authors have declared that no competing interests exist.

## Introduction

Species are fundamental evolutionary units that are separated from each other via different prezygotic and postzygotic barriers [1]. Such barriers prevent gene flow between different species, ensuring the isolation of their genomes [1, 2]. Interspecific hybridisation occurs both in plants and animals, causing the instant creation of novel gene combinations [3, 4]. These novelties may promote the evolutionary success of hybrid lineages, and possibly, the emergence of new species [3–6]. However, one of the outputs of interspecific hybridisation is hybrid sterility, which prevents recombination and thus, gene flow between the genomes of parental species [7, 8]. In some animal hybrids, sterility is often overcome via modifications of gametogenesis by the emergence of clonal or hemiclonal reproduction [9–11]. These forms of reproduction become possible when gamete formation is modified to prevent pairing thus, preventing recombination between orthologous chromosomes [11, 12]. Clonal propagation of the genome usually leads to the emergence of triploid individuals with the frequent ability to fully or partially restore recombination (sexual reproduction) [11–15]. This phenomenon of asexuality accompanied by polyploidisation may involve multiple transient stages in which different forms of hybrids reproduce with parental (sexual) species or with each other [14, 16]. However, how these stages of hybrid reproduction are manifested and what mechanisms provide maintenance of hybrids remains unresolved.

The water frog complex (*Pelophylax esculentus* complex) provides several opportunities to study hemiclonal reproduction and polyploidisation. This complex includes two parental species, namely, the marsh frog (*P. ridibundus*; RR genotype, 2n = 26) and the pool frog (*P. lessonae*; LL genotype, 2n = 26), the hybridisation of which leads to diploid hybrids, i.e., the edible frog (*P. esculentus*; RL, 2n = 26) [17]. In addition to diploid hybrids, triploid (RRL and LLR, 3n = 39), tetraploid (RRLL), and rare pentaploid (RRLLL) hybrids have been recorded in natural populations [18–21]. Diploid hybrids exhibit a hemiclonal reproductive mode, known as hybridogenesis [22]. During gametogenesis, the genome of one parental species is eliminated, while the other genome is duplicated, allowing for its transmission into gametes [22–24]. In restoring the next hybrid generation, such gametogenic properties create hybrids depending on one of the parental species. This leads to the formation of various populations where hybrids coexist with both parental species (R-E-L systems) and one or another parental species (L-E or R-E systems, respectively) [18, 19, 25]. Alteration of diploid hybrid gametogenesis leads to the appearance of fertile triploid hybrids, which partially restores sexual reproduction [14, 18, 26]. During gametogenesis of triploid hybrids, the genome represented in one copy is typically eliminated, while the others recombine to form haploid gametes [18, 19, 26, 27]. Diploid and triploid hybrids can coexist independently from parental species, producing self-maintaining evolutionary units (known as "E systems") [18, 19, 25, 28, 29]. Such populations are widespread in Sweden, Denmark, Germany, Poland, and Ukraine [28–32]. Here, we focused on the reproduction of hybrid frogs in pure hybrid populations to determine contribution of hybrid animals to the progeny.

In this study, we considered whether different pure hybrid systems have similar hybrid reproduction patterns. To investigate this, we selected two E systems located in northwest (Wysoka Kamieńska) and southwest (Uciechów) Poland. In the E and L-E systems observed in Wysoka Kamieńska, all three forms of edible frog (RL, LLR, LRR) co-occur, with the predominant form being LLR hybrids (approximately 60% of frogs per population) [33]. This population in Wysoka Kamieńska is closely located to other E systems in Usedom and Bornholm Islands, northern Germany, Denmark, and southern Sweden, which have been comprehensively studied [28–30, 32, 34, 35]. Diploid hybrids from studied E systems mainly produce haploid *P. ridibundus* gametes and diploid gametes, leading to the emergence of triploid hybrids

[20, 28, 29, 34, 35]. Triploid hybrids with the LLR genotype usually produce haploid gametes with the *P. lessonae* genome, as well as diploid gametes with *P. ridibundus* and *P. lessonae* genomes and gametes with two copies of the *P. lessonae* genome [27, 28, 32–34]. The other selected E system, located in southwest (Uciechów) Poland, is remote from the northern systems and likely represents a separate case of pure hybrid population formation. According to our previous unpublished long-term field observations, this population included only hybrid animals and was isolated from other water frog populations. However, the genotype composition of individuals from Uciechów has not been described thus far; therefore, we aimed to characterise the structure of this population and the pattern of hybrid emergence and maintenance in this E system.

Finally, we investigated whether all types of gametes produced by hybrids contributed to the progeny genome composition. We crossed the same hybrid individuals with both parental (sexual) species, and other hybrids, to compare the genotypes of their progeny. Previous studies have reported that hybrids can simultaneously produce a variety of gametes with different genome compositions and ploidy [28, 29, 35–39]. We expected to find no difference in the contribution of hybrid gametes to the emergence of tadpoles with different genome combinations after crosses with different parents. To test this, we performed a karyotype analysis of tadpoles obtained after artificial crosses of di- and triploid hybrid frogs with each other and with parental species from different populations. We identified karyotypes of 655 tadpoles from 27 crosses of hybrid frogs via fluorescent *in situ* hybridisation (FISH) using species-specific markers. The results allowed us to determine the contribution of hybrid frogs to the appearance of different forms of hybrids in E systems and the role of postzygotic barriers on the survival and maintenance of the progeny.

## Materials and methods

### Animals

All procedures with both adults and tadpoles were performed in accordance with the relevant guidelines and regulations. Adult frogs were captured by the General and Regional Directorates for Environmental Protection (DZP-WG.6401.02.5.2015.JRO, WPN.6401.177.2016.IL). Adult individuals used for *in vitro* crosses were sampled from different locations in Poland. According to the data collected in S1 Table, *P. lessonae* (females, N = 5; males, N = 8) were obtained from the L-E and R-E-L systems; *P. ridibundus* (females, N = 1; males, N = 6) were collected from R-E and R-E-L systems. *P. esculentus* (RL females, N = 9; LLR females, N = 3; RL males, N = 9; LLR males, N = 3; RRL males, N = 1) were taken from E systems in Wysoka Kamieńska (northwestern Poland), Uciechów (southwestern Poland), and other locations. All experimental procedures were approved by the Local Commission for Ethics in Experiments on Animals in Wrocław, Poland (27/2016) and the local animal ethics committee of Saint-Petersburg State University (# 131-03-3).

### Taxonomic evaluation of animals

Adult frogs collected for crossing experiments were first identified based on morphological features according to Kierzkowski et al. [40]. Ploidy of *P. esculentus* individuals was initially evaluated via erythrocyte long-axis measurements before crosses [40]. Phalanges were carefully cut to obtain blood smears. After air-drying the erythrocytes, the long axis was measured under an Axiostar Plus microscope (Zeiss) using 20× lenses and KS400 software (Zeiss). Diploid erythrocytes were 23.4–24.9 μm long, and triploid erythrocytes were 29.5–33.3 μm long. After crossing experiments, all frogs were identified by karyotyping followed by FISH with species-specific markers, such as pericentromeric repeat RrS1 [41–43] and interstitial sites of the telomeric repeat (ITSs) [36, 44].

## Crossing experiments

The collected animals were kept in humid boxes and subsequently used for *in vitro* crossing experiments according to the standard procedures for water frogs [45]. Twenty-four hours before the procedure, females were injected intraperitoneally with 6.25 mg/kg-body weight of salmon luteinising hormone-releasing hormone (LHRH, H-7525.0001, Bachem) in amphibian phosphate-buffered saline (APBS, pH 7.4, 11.2 mM NaCl, 0.22 mM KCl, 0.8 mM $Na_2HPO_4$, 0.14 mM $KH_2PO_4$). Eggs were artificially obtained from each female and separated into several plates (depending on the clutch size) for fertilisation by different males. Males and females were sacrificed after anaesthetising in a 0.5% solution of ethyl 3-aminobenzoate methanesulfonate (tricaine methanesulfonate, MS-222, Sigma Chemical Co.) in APBS. Tissues and organs were dissected and fixed in 96% ethanol for subsequent DNA isolation (phalange) and chromosomal (intestine) analyses, as well as for *in vitro* fertilisation of eggs (testes). Tadpoles of various phenotypes (LL, RR, RL, RRL, LLR), resulting from controlled *in vitro* crosses, were reared in a greenhouse in PPE tanks, with approximately 10 tadpoles per litre of water and fed with frozen lettuce and fish food. A total of 655 tadpoles were analysed from 27 crosses of hybrids with each other and parental species.

Tadpole genome composition was identified by karyotyping followed by FISH using species-specific probes. Using the (TTAGGG)$_5$ probe, we revealed one or two interstitial telomere repeat sites distinguishing the NOR-bearing chromosomes of *P. lessonae* and *P. ridibundus* [36, 44]. In addition, we applied a probe to detect the centromere RrS1 repeat, which only localises in *P. ridibundus* chromosomes [41–43].

## Metaphase chromosome preparation

Tadpoles were anaesthetised with 0.15% MS222 (Sigma) before euthanasia. The gills, intestine, and tail tip were hypotonised in distilled water for 30 min, followed by fixation in a 3:1 (ethanol:glacial acetic acid) Carnoy's solution. After three fixative exchanges, tissues were stored at 4°C until use. To obtain metaphase chromosomes, fixed tissues were placed in 70% glacial acetic acid for 3 min and intensively macerated to obtain a cell suspension. The suspension was dropped onto slides heated to 60°C, on which nuclei and metaphase chromosomes remained after liquid evaporation.

## FISH

To identify chromosomes of both parental species, we applied DNA/DNA FISH protocols with an oligonucleotide probe (TTAGGG)$_n$ repeat or a PCR-labelled probe to RrS1 repeat on metaphase chromosomes obtained from tadpoles and parental individuals. A biotin-labelled probe for RrS1 pericentromeric repeat was obtained from the genomic DNA of *P. ridibundus* using PCR with the following primers (according to [41]): 5′-AAGCCGATTTTAGACAAGAT TGC-3′; 5′-GGCCTTTGGTTACCAAATGC-3′.

Metaphase chromosomes were pre-treated with RNAse A (100–200 mg/ml) for 1 h and pepsin (0.01% in 0.01 N HCl) for 10 min and then post-fixed in 2% paraformaldehyde for 10 min (in 1× PBS, 50 mM $MgCl_2$). For the oligonucleotide probe, the hybridisation mix contained 40% formamide, 2.4× SSC, 12% dextran sulphate, 5 ng/µl single-stranded (TTAGGG)$_5$ probe conjugated with biotin and 10–50-fold excess of tRNA. For the PCR-labelled probe, the hybridisation mix contained 50% formamide, 2× SSC, 10% dextran sulphate, 5 ng/µL probe conjugated with biotin, and 10–50-fold excess of ssDNA. The hybridisation mix was applied to slides with metaphase chromosomes under coverslips. Slides with chromosomes and probes were denatured simultaneously for 5 min at 82°C. For the oligonucleotide probe, hybridisation was performed overnight at room temperature, followed by washing in 2× SSC at 42°C.

Regarding the PCR-labelled probe, hybridisation was performed overnight at 37˚C, followed by washing in 0.2× SSC at 50˚C. Biotin-labelled probes were detected using streptavidin conjugated with Cy3 (Jackson ImmunoResearch Laboratories) or Alexa 488 (Invitrogen). Subsequently, the slides were counterstained in 1,4-Diazabicyclo[2.2.2]octane (DABCO) antifade solution containing 1 mg/ml 4′,6-diamidino-2-phenylindole (DAPI). At least three full metaphase plates with evident FISH signals were examined to identify the karyotypes of the tadpoles and their parental individuals.

## Wide-field microscopy

A Leica fluorescence microscope DM 4000B was used to analyse metaphase chromosomes. Fluorescent signals were detected using appropriate filter cubes (Leica Wetzlar GmbH, Germany). Images were taken using a monochrome digital camera DFC350 FX under 10×, 20×, 40×, and 100× objective lens magnifications using the Leica CW 4000 FISH software. Proper adjustments of the images were performed using Adobe Photoshop and Adobe Illustrator.

## Results

We analysed 33 hybrid individuals from two isolated pure hybrid *P. esculentus* systems (E systems) located in southwest (Uciechów) and northwest (Wysoka Kamieńska) Poland. Both populations shared similar genotypes. Males were represented by all three genotypes (RL, RRL, LLR), while females were represented by RL and LLR genotypes; RRL females were not found (S1 Table).

### Crosses of hybrid frogs from E-system from Uciechów

To investigate the mechanisms of hybrid frog maintenance in a previously undescribed E system located in southwestern Poland (Uciechów), we performed a set of crossing experiments of hybrid frogs with the other hybrids and with the parental species. We performed 17 crosses of one triploid and eight diploid hybrid females with *P. ridibundus*, *P. lessonae*, as well as triploid hybrid males with LLR and RRL genotypes (Fig 1; S2 Table). In addition, we crossed *P. lessonae* females and two diploid hybrid males.

In one cross of a diploid hybrid female with a *P. lessonae* male (cross # 37/2016), we obtained only diploid hybrid tadpoles (Figs 1, 3C; S2 Table). In crosses of two other diploid hybrid females with *P. ridibundus* males (crosses ## 14/2016, 18/2016), we obtained only *P. ridibundus* tadpoles (Fig 1, S1G Fig; S2 Table). Thus, these females produced eggs with only the *P. ridibundus* genome.

To check whether the hybrid females' contribution to the progeny was comparable to crosses with different parents, we split the eggs obtained from hybrid individuals and fertilised them with the sperm of various sexual males. In crosses of diploid hybrid female with *P. ridibundus* (cross # 15/2016) and *P. lessonae* (cross # 16/2016) males, we obtained *P. ridibundus* and *P. esculentus* tadpoles, respectively (Figs 1, 3L; S1K Fig; S2 Table). We concluded that the studied females transmitted oocytes only with the *P. ridibundus* genome (Fig 1).

In addition, we crossed another diploid hybrid female with a *P. lessonae* male (cross # 2/2016), a *P. ridibundus* male (cross # 7/2016), and triploid hybrid male with the LLR genotype (cross # 5/2016) (Figs 1, 3F–3I; S1M and S1N Fig). In crosses from this female with *P. lessonae* and triploid hybrids, we obtained only diploid hybrid progeny. However, in a cross with *P. ridibundus*, we detected not only *P. ridibundus* progeny, but also diploid hybrids. We concluded that this female simultaneously produced haploid gametes with the *P. ridibundus* genome and haploid gametes with the *P. lessonae* genome (Fig 1).

| Crosses' ID | Females genotypes | Suggested females' gametes | Males genotypes | Suggested males' gametes | Tadpoles genotypes | Number of analyzed tadpoles |
|---|---|---|---|---|---|---|
| 37/2016 | RL | R | LL | L | RL | 24 |
| 14/2016 | RL | R | RR | R | RR | 25 |
| 18/2016 | | | | | | 24 |
| 15/2016 | RL | R | RR | R | RR | 25 |
| 16/2016 | | | LL | L | RL | 34 |
| 02/2016 | RL | R | LL | L | RL | 26 |
| 07/2016 | | L | RR | R | RR | 19 |
| | | | | | RL | 9 |
| 05/2016 | | | LLR | L | RL | 20 |
| 03/2016 | RL | RL | LL | L | LLR | 6 |
| | | | | | RL | 22 |
| 08/2016 | | R | RR | R | RRL | 5 |
| | | L | | | RR | 16 |
| | | | | | RL | 8 |
| 07/2017 | RL | R | LL | L | LLR | 2 |
| | | RL | | | RL | 8 |
| 09/2017 | | ? | RL | R | RL | 4 |
| | | | | ? | RR | 11 |
| 11/2016 | RL | R | LL | L | RL | 29 |
| | | ? | | | | |
| 12/2016 | | | RRL | R | LLR | 4 |
| | | | | | RRL | 4 |
| | | | | ? | RL | 11 |
| | | | | | RR | 8 |
| 08/2017 | LLR | R | LL | L | RL | 4 |
| | | L | | | LL | 11 |
| 10/2017 | | | RL | R | RR | 3 |
| | | | | | RL | 16 |
| 15/2017 | | | RR | R | RL | 11 |
| 09/2016 | LL | L | RL | R | RL | 36 |
| 17/2017 | LL | L | RL | R | RL | 14 |
| | | | | L | LL | 3 |

**Fig 1. Results of crossing experiments of diploid and triploid hybrids from pure hybrid system located in southwest (Uciechów) Poland.** Genome composition of tadpoles was identified by karyotyping followed by FISH with probes to centromeric repeat RrS1 and telomeric $(TTAGGG)_n$ sequence. RR indicates *P. ridibundus* individuals; LL indicates *P. lessonae* individuals; RL indicates diploid hybrids; LLR and RRL indicate triploid hybrid individuals. Suggested genome composition of gametes produced by males and females is inferred based on tadpole karyotypes and parent genotypes. R and L represent gametes with *P. ridibundus* and *P. lessonae* genomes, respectively. RL indicates diploid gametes with the genomes from both parental species. Question mark (?) indicates difficulties in prediction of the precise genome composition in gametes produced by hybrid males or females. Crossing IDs correspond to S2 Table.

Furthermore, we crossed another diploid hybrid female with *P. lessonae* (cross # 3/2016) and *P. ridibundus* (cross # 8/2016) males (Fig 1). In the cross with the *P. lessonae* male (cross # 3/2016), we obtained diploid and triploid hybrids with the LLR genome (Fig 1; S1E, S1F Fig). When we crossed this female with the *P. ridibundus* male, we obtained *P. ridibundus* and triploid hybrids with the RRL genotype (cross # 8/2016), as expected, and diploid hybrids (Figs 1 and 3M–3O). Thus, this diploid hybrid female simultaneously produced three types of gametes: the majority of the gametes were haploid with the *P. ridibundus* genome, and a minor proportion of gametes was haploid with the *P. lessonae* genome or diploid with genomes of both parental species (Fig 1).

In crosses of an additional diploid hybrid female with a *P. lessonae* male (cross # 7/2017), we obtained triploid hybrids with the LLR genotype and diploid hybrids (Figs 1, 3G and 3H). However, when we crossed the same female with diploid hybrid male (cross # 9/2017), all progenies were diploid, including the *P. ridibundus* and *P. esculentus* individuals (Figs 1, 3J and 3K). We concluded that this diploid hybrid female produced diploid eggs with genomes of both parental species, as well as haploid eggs with the *P. ridibundus* genome. The diploid hybrid male presumably produced two types of haploid gametes with the *P. ridibundus* genome (Fig 1). Nevertheless, we could not infer the genome composition of all gametes produced by hybrid females and hybrid males, as it is possible for hybrid males and hybrid females to produce gametes with the *P. lessonae* genome.

In a cross of another diploid hybrid female and a *P. lessonae* male (cross # 11/2016), we obtained only diploid hybrid tadpoles (Fig 1, S1L Fig). We suggest that this diploid hybrid female produced haploid gametes with the *P. ridibundus* genome. However, when we crossed the same female and triploid hybrid males with the RRL genotype (cross # 12/16), we obtained triploid hybrids with RRL and LLR genotypes, as well as diploid hybrids and *P. ridibundus* individuals (Fig 1, S1A–S1D Fig). We cannot precisely identify the additional gametes of the crossed hybrids, but we can infer the formation of diploid eggs with genomes of both parental species and haploid eggs with the *P. lessonae* genome, which were undetected in the crosses of the same female with *P. lessonae* males (Fig 1). Alternatively, triploid hybrid males with the RRL genotype can produce spermatozoa with four different genotypes (diploid RL, diploid LL, haploid R, and L); however, this scenario is less probable.

Next, we crossed one triploid LLR female with *P. lessonae* male (cross # 8/2017), diploid hybrid male (cross # 10/2017), and *P. ridibundus* male (cross # 15/2017) (Fig 1, S2A, S2B, S2D–S2F Fig). After crossing this female with *P. ridibundus* male, we found only diploid *P. esculentus* tadpoles. However, when we crossed the same female with *P. lessonae* male, we obtained *P. esculentus* and *P. lessonae* tadpoles. *P. ridibundus* and *P. esculentus* tadpoles were obtained from a cross of the same female with *P. esculentus* male. These results allowed us to speculate that this triploid hybrid female produced haploid oocytes with *P. lessonae* or *P. ridibundus* genomes (Fig 1).

In addition, we crossed two diploid hybrid males with *P. lessonae* females. In one cross (cross # 9/2016), we detected only diploid hybrid tadpoles (Fig 1). We concluded that this

diploid hybrid male produced gametes with the *P. ridibundus* genome. However, in the second cross (cross # 17/2017), we found diploid hybrids and *P. lessonae* individuals (Fig 1; S2H and S2I Fig). We suggest that this individual male simultaneously produced two types of gametes: haploid gametes with the *P. ridibundus* genome and haploid gametes with the *P. lessonae* genome.

### Results of hybrid frog crosses from Wysoka Kamieńska

Additionally, we analysed the mechanisms of hybrid frog maintenance in different E systems located in northwest Poland (Wysoka Kamieńska). We performed eight crosses of one diploid and two triploid LLR hybrid females with hybrid males and with males of parental species. In addition, we crossed five diploid and two triploid LLR males with parental species (S2 Table).

We performed crosses of one diploid hybrid female with *P. lessonae* male (cross # 1/2016) and triploid hybrid male with LLR genotype (cross # 4/2016). In both crosses, we obtained diploid and triploid LLR hybrid tadpoles (Figs 2, 3A, 3B, 3D and 3E). We suggest that this diploid hybrid female transmitted oocytes with the haploid *P. ridibundus* genome and diploid oocytes with genomes of both parental species; triploid hybrid males likely produced gametes with the *P. lessonae* genome (Fig 2). Nevertheless, triploid hybrid male was able to also produce diploid sperm with genomes of both parental species (Fig 2).

In addition, we obtained tadpoles from two crosses of two triploid hybrid females with the LLR genotype and diploid hybrid males. In one cross (cross # 26/2016), we found tadpoles of *P. ridibundus*, diploid hybrids, and triploid hybrids with the LLR genotype (Fig 2; S2I–S2K Fig). In this case, we could not identify all types of gametes produced by hybrid parents, but we concluded that both male and female hybrids produced *P. ridibundus* gametes, and one of the parents produced diploid gametes. In another cross of triploid hybrid female with the LLR genotype and diploid hybrid male (Cross # 27/2016), we obtained *P. ridibundus* and diploid hybrid progeny (Fig 2; S2G and S2H Fig). Similar to the previously described cross, we could not precisely identify the gametes produced by hybrid parents, but we suggest that both male and female hybrids produced haploid gametes with the *P. ridibundus* genome, and at least one of the parents additionally produced haploid gametes with the *P. lessonae* genome (Fig 2).

Next, we performed three crosses of diploid hybrid males with *P. lessonae* females. In two crosses of hybrid males (crosses # 23/2016 and # 28/2016), we detected only diploid hybrid tadpoles (Fig 2; S2M Fig). We concluded that diploid hybrid males from the studied crosses produced gametes with the *P. ridibundus* genome (Fig 2). All tadpoles from one cross of hybrid male and *P. lessonae* female (cross # 31/2016) were *P. lessonae* (Fig 2; S2C Fig). Therefore, the diploid hybrid male produced gametes with the *P. lessonae* genome (Fig 2).

Additionally, we analysed tadpoles from a crossing of *P. ridibundus* female with triploid hybrid male with the LLR genotype (cross # 30/2016). All analysed tadpoles were diploid *P. esculentus* (Fig 2; S2L Fig). Therefore, this triploid hybrid male produced gametes with the *P. lessonae* genome (Fig 2).

## Discussion

### Gamete formation in hybrids from the studied pure hybridogenetic water frog populations

The two studied pure E systems located in northwest (Wysoka Kamieńska) and southwest (Uciechów) Poland have similar structures and genetic compositions. In both populations, we detected diploid RL and triploid RRL and LLR hybrids, with predominant LLR genotypes (S1 Table). This result corresponds with previous data obtained for other E systems from

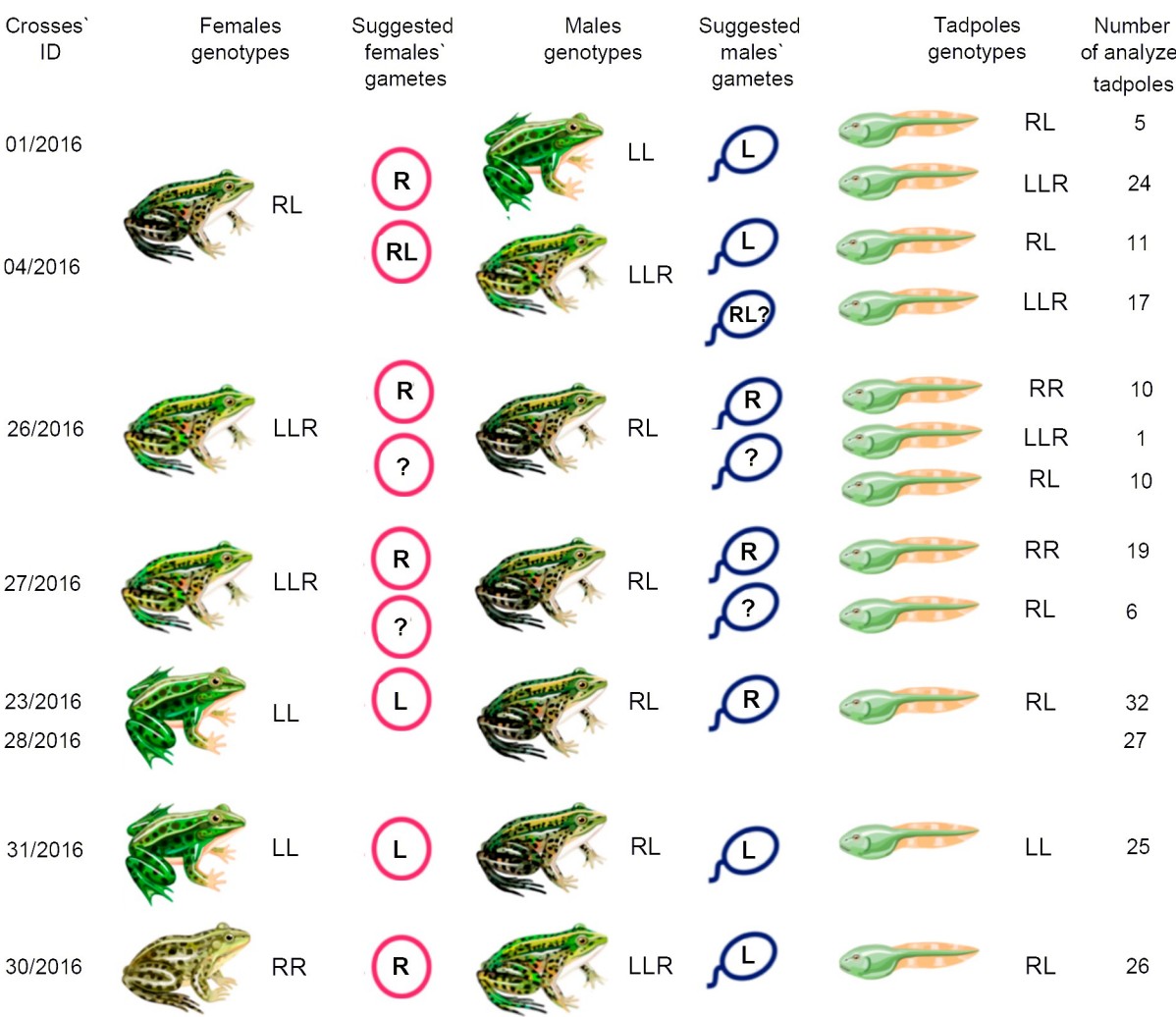

**Fig 2. Results of crossing experiments of diploid and triploid hybrids from pure hybrid system located in the northwest (Wysoka Kamieńska) Poland.** Genome composition of tadpoles was identified by karyotyping followed by FISH. RR indicates *P. ridibundus* individuals; LL indicates *P. lessonae* individuals; RL indicates diploid hybrids; LLR and RRL indicate triploid hybrid individuals. Suggested genome composition of eggs and sperm is inferred based on tadpoles' karyotypes and parents'genotypes. R and L indicate gametes with *P. ridibundus* and *P. lessonae* genomes, respectively. RL indicates diploid gametes with the genomes from both parental species. Question mark (?) shows inability to deduce precise genome composition of gametes produced by hybrid males or females. Crossing IDs correspond to S2 Table.

northwest Poland, northern Germany, Denmark, and southern Sweden [28–30, 32, 34, 35]. Moreover, our results are strongly corroborated by early observations performed on E systems in northwest (Wysoka Kamieńska) Poland [33, 34]. Therefore, we propose that the water frog genotype composition is highly stable in E systems, and gametes produced by hybrids of different genotypes and ploidy are sustainable.

The genome composition of tadpoles obtained in a series of artificial crosses allowed us to establish the varieties of gametes produced by diploid and triploid hybrids. Moreover, we observed similarities between the mechanisms of hybrid frog maintenance in the two isolated pure hybrid systems. Diploid hybrid females from both analysed E systems predominantly produced gametes with *P. ridibundus* genomes and diploid gametes with both parental genomes (Fig 4). Diploid gametes are crucial for the formation of triploid hybrids with both

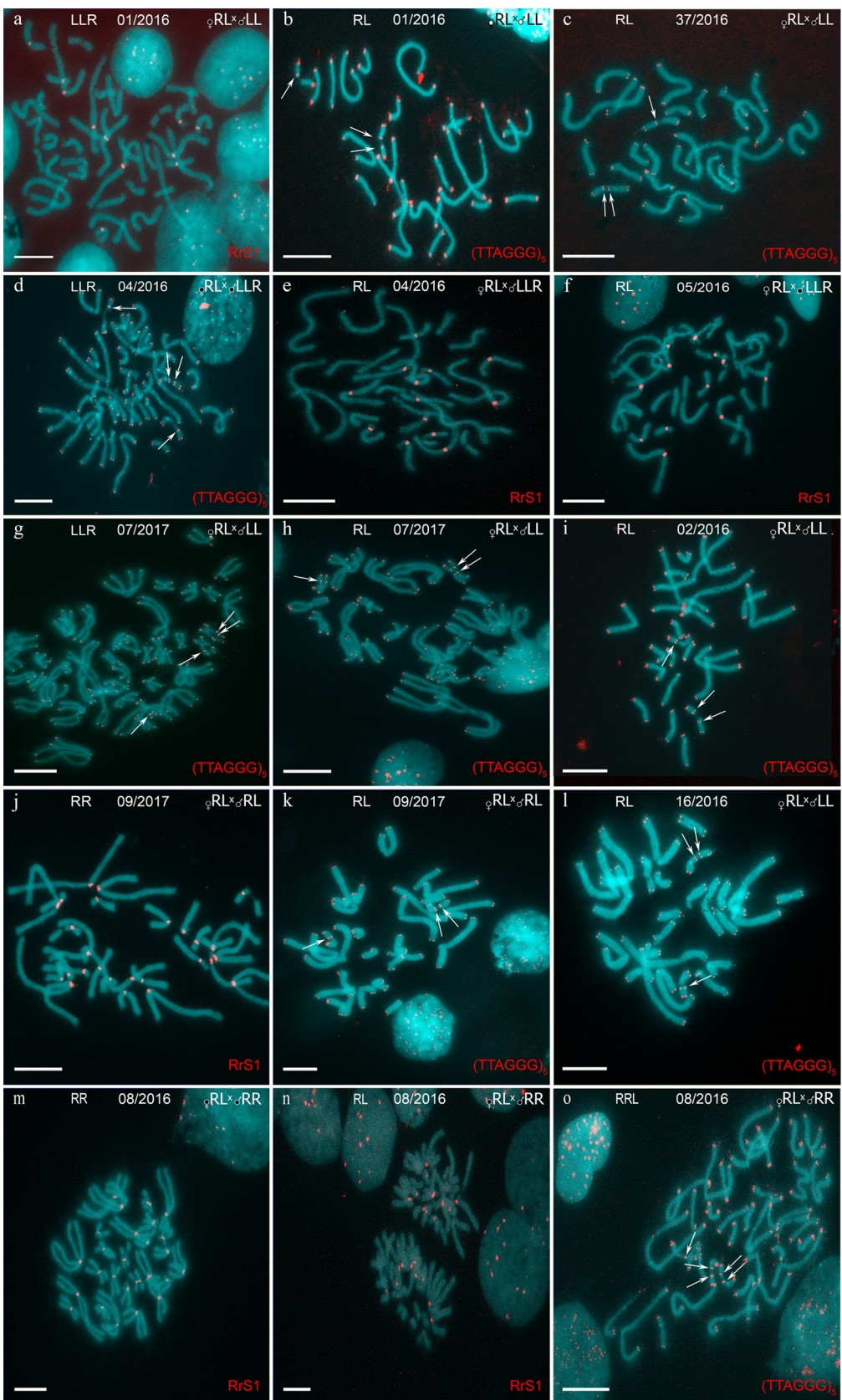

**Fig 3.** Identification of the tadpoles from crossings of diploid hybrid females with *P. lessonae* males (a-c, g-i, l), *P. ridibundus* males (m-o), diploid hybrid males (j, k), and triploid hybrid males with LLR genotype (d-f). Metaphase chromosomes from tadpoles after FISH with (TTAGGG)$_5$ (b-d,g-i,k,l,o). One or two interstitial (TTAGGG)$_n$ repeat sites were distinguished in *P. lessonae* and *P. ridibundus* NOR-bearing chromosomes, respectively. Arrows indicate interstitial (TTAGGG)$_n$ repeat sites. FISH with the RrS1 (a,e,f,j,m,n) probe allows distinguishing pericentromeric regions of only *P. ridibundus* chromosomes. According to karyotype and species-specific FISH markers, we distinguished tadpoles of *P. ridibundus* (j,m), diploid hybrids (b,c,e,f,h,i,k,l,n), and triploid hybrids with LLR (a,d,g) and RRL (o) genome compositions. Crossing IDs correspond to Figs 1, 2 and S2 Table. Scale bars = 10 μm.

the LLR and RRL genotypes. The role of diploid hybrid females in the formation of triploid progeny has been shown in a number of other E systems [14, 20, 28, 29, 33], as well as in L-E and R-E systems [17, 18, 36, 37, 42, 46]. We also found that at least one diploid hybrid female from Uciechów produced haploid gametes with the *P. lessonae* genome along with diploid and haploid gametes with *P. ridibundus* genomes. Such gametes have not been found in other E systems [20, 25, 28, 29, 34, 35, 47–49].

Diploid *P. esculentus* males mostly produced gametes with the *P. ridibundus* genome. Nevertheless, in both populations, we detected rare diploid hybrid males producing gametes with only the *P. lessonae* genome or gametes with a mixture of the *P. lessonae* and the *P. ridibundus* genomes. Males, which simultaneously produce two types of gametes, have been previously observed in other systems and are particularly abundant in R-E systems [37, 38, 50–52]. These results were obtained by measuring sperm genome DNA using flow cytometry [38, 51] and determined after an analysis of progeny [37], as well as cytogenetic analysis of meiosis [50]. The simultaneous formation of two types of gametes can be explained by alterations in genome elimination/endoreplication in different germ cell lines [36, 51].

Triploid hybrid males with the LLR genotype produced haploid sperm with the *P. lessonae* genome in both analysed populations in Wysoka Kamieńska and Uciechów. Our results are in agreement with earlier reports that triploid hybrids usually eliminate the genome that presents as a single copy (LLR hybrids eliminate R genome; RRL hybrids eliminate L genome) and transmit a two-copy genome to haploid gametes after normal meiosis [14, 18–20, 26, 28, 36]. In contrast to triploid hybrid males, triploid hybrid females with the LLR genotype

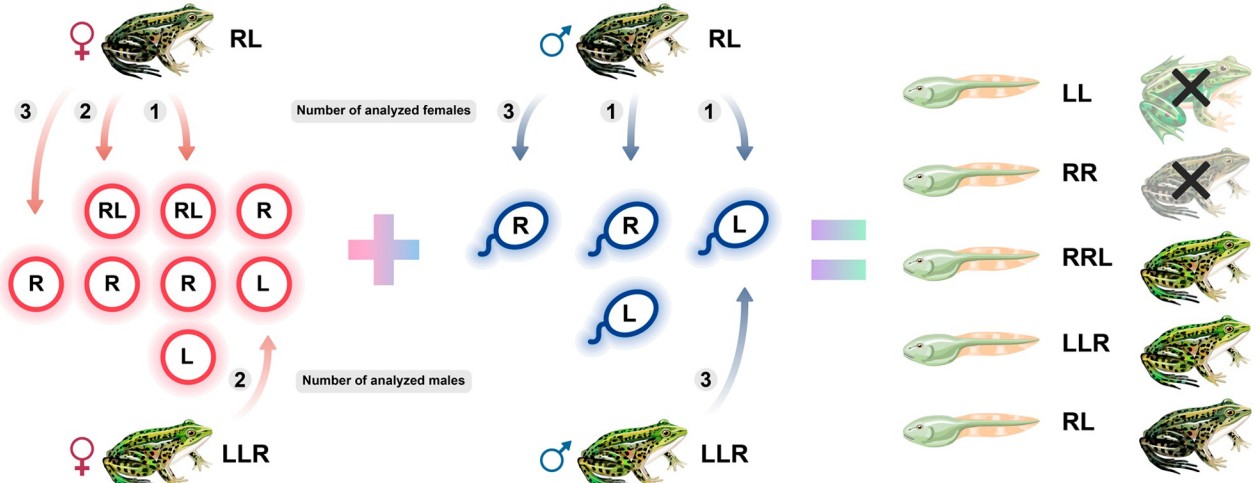

**Fig 4. Schematic overview of contribution of diploid and triploid water frog hybrids to the maintenance of pure hybrid systems.** Data from both the studied pure hybrid systems are summarised with the indication of genotypes, sex, and gametes of hybrid frogs, tadpole genotypes, as well as presumptive adult progeny. Female gametes are indicated in pink, and male gametes are indicated in blue. Gametes produced by RRL individuals were not included in the schematic because of the low number of investigated individuals.

simultaneously produced a mixture of gametes with *P. lessonae* and *P. ridibundus* genomes. The simultaneous formation of gametes with various genome compositions has been shown for rare triploid LLR females from E systems [28, 29].

Diploid and triploid hybrids in both populations produced similar types of gametes, suggesting a common pattern of emergence and maintenance of pure hybrid populations. Hybrid males and females of both ploidy levels in the studied E systems produced gametes with various genome compositions and ploidy levels, suggesting different contributions to the progeny genotypes. Based on our results, we suggest a pattern of hybrid emergence in the studied E systems from Poland (Fig 4). The dominant fraction of gametes produced by diploid males and females are those with the *P. ridibundus* genome; gametes with the *P. lessonae* genome are mostly produced by triploid LLR hybrids of both sexes. The suggested reproductive patterns are similar to those previously proposed for hybrids from the other E-type systems located in Denmark and southern Sweden [14, 18–20, 28, 29]. Hence, the processes underlying the emergence and maintenance of hybrids are essentially the same in the E systems studied thus far.

## Impact of hybrid gametogenesis on the survival rates in different crosses

We found high variability of progeny genotypes when crossing one female with several males, which indicates different survival rates of progeny depending on parental genotypes. Previous studies have shown that hybrid frogs simultaneously produce gametes that differ in ploidy and genotypes [14, 28, 29, 36, 37, 39, 42, 46]. In these studies, the genomic composition of gametes was inferred by investigating the genome composition of oocytes and identifying the karyotypes of progeny [14, 28, 29, 36, 37, 39, 42, 46]. However, here we found that not all of the gametes produced were able to contribute to viable tadpoles if they were fertilised by sperm from different parental species or hybrid males. According to our results, three diploid (43%) and two triploid (50%) *P. esculentus* females were able to simultaneously produce gametes with different ploidy levels and genotype composition. We also detected decreased survival of tadpoles with parental genotypes, which appeared after crossing hybrids with different parental species. In two crosses of diploid and triploid hybrid females with the LLR genotype, we were not able to detect *P. lessonae* progeny, whereas *P. esculentus* tadpoles were detected if the same females were crossed with *P. ridibundus* males. In the cross of triploid hybrid female and *P. ridibundus* male, we did not detect *P. ridibundus* tadpoles, wherein *P. esculentus* tadpoles were documented in the cross of the same female and *P. lessonae* male. The decreased survival rate of neo-parental species progeny can be explained by deleterious mutations accumulated in the clonally transmitted *P. ridibundus* and *P. lessonae* genomes [53–55]. When eggs containing the clonal *P. lessonae* genome are fertilised by hybrid sperm bearing the *P. lessonae* genome, it leads to the death of *P. lessonae* or *P. ridibundus* embryos [17, 54–56]. Earlier, it was shown that *P. ridibundus*, which emerged from the crosses of two hybrids transmitting the same genome from L-E systems, has multiple developmental abnormalities and is ultimately not viable [17, 46, 57]. Similarly, *P. esculentus* males from R-E systems in central Europe often simultaneously produce two types of gametes with *P. ridibundus* and *P. lessonae* genomes based on the identification of the tadpole genotypes [52]. After crosses with *P. ridibundus* females, the majority of progeny consisted of *P. esculentus* males, suggesting that *P. ridibundus* individuals likely die during early development [52]. In the studied E systems from northwestern and southwestern Poland, we did not find adult *P. ridibundus* and *P. lessonae* individuals, despite the detection of *P. ridibundus* and *P. lessonae* tadpoles, even after crosses of two hybrids from the same locality. These results suggest that parental species have decreased survival rates at later developmental stages and are likely to die during metamorphosis. Moreover, parental species may have different fitness values than hybrid individuals [58]. Thus, the viability of the

certain progeny genotypes indicates the strong effect of postzygotic rather than prezygotic barriers on the maintenance of studied hybrid systems. Comparative analysis of crosses results and hybrids gametogenesis in E [28, 29, 59], R-E [36–38, 50, 51, 60, 61], L-E [17, 39, 42, 54, 56, 62, 63] and R-L-E [42] systems, and direct observations of eliminated genome in hybrid tadpoles [42, 43, 64] clearly show that *P. lessonae* genome is preferentially eliminated in contrast to *P. ridibundus* genome. Nevertheless, striking variability of gametes produced by diploid and triploid hybrids in E systems suggest aberrancies in the genome elimination and endoreplication during hybrids frog gametogenesis (this study, [28, 29, 59]). Such aberrancies may be caused by the genetic introgression from other water frog species into the genetic background of *P. ridibundus* and thus completely or partially prevent hybridogenetic reproduction [63]. Additionally, the ability of *P. lessonae* genome to persist elimination can cause competition between different genomes to be eliminated or retained, resulting in gamete variability in one individual. Interestingly, that R-E systems in the central Europe, in which diploid hybrid males produce two types of sperm, have a single origin suggesting the possibility of latter scenario [50, 52]. Nevertheless, the mechanism of genome elimination in water frogs requires further investigation.

Based on the high variability of gametes produced by diploid and triploid hybrids, as well as the various survival rates of the resulting tadpoles, we suggest that not all hybrids in E systems follow the "classical" hybridogenetic pathways. The genome variability of gametes produced by hybrid individuals from E systems is significantly higher than that in hybrids from L-E systems, which exhibit a stable and transient formation of gametes [17, 39, 42, 57, 63]. The variability of gametes produced by hybrids likely increases the number of possible combinations of genotypes in their progeny. Moreover, the survival rate of particular tadpole genotypes also depends on the genotype of the parents involved in a particular cross.

## Conclusion

By cytogenetic analysis of tadpoles obtained from various crosses of hybrid frogs from two separate pure hybrid systems in Poland, we characterised the hybrid contribution to the emergence and maintenance of hybrids in populations without parental species. We found similarities in gamete formation between hybrids in the two separate E systems. Diploid hybrid males primarily produced gametes with the *P. ridibundus* genome or a mixture of haploid gametes with *P. ridibundus* and *P. lessonae* genomes; diploid hybrid females produced gametes with the *P. ridibundus* genome and a mixture of haploid gametes with the *P. ridibundus* genome, as well as diploid gametes with genomes of both parental species. Triploid hybrid males primarily produced haploid gametes with the *P. lessonae* genome, while triploid hybrid females produced a mixture of haploid gametes with *P. ridibundus* and *P. lessonae* genomes. The majority of diploid and triploid hybrids simultaneously produced several types of gametes, which differed in genome composition and ploidy level. Moreover, we observed sex-specific differences in the contribution to the progeny in both diploid and triploid hybrids. In addition, we detected preferential variability in the survival rate of particular genotypes, depending on which species the progeny is crossed with, suggesting that postzygotic barriers play an important role on the maintenance of hybrid systems. Such dependence of the variability in progeny genotypes on the genotypes of their parents has crucial methodological consequences, as progeny genotypes can be biased based on the survival rate of a particular genotype.

## Supporting information

**S1 Fig.** Identification of the tadpoles from crossings of diploid hybrid females with *P. lessonae* males (e,f,l,m), *P. ridibundus* males (g,k,n), diploid hybrid males (h,i), and triploid hybrid

males with LLR genotype (a-d, j). Metaphase chromosomes from tadpoles after FISH with (TTAGGG)$_5$ (a-f,h,i) and RrS1 (g,j,k,n) probes. Arrows indicate interstitial (TTAGGG)$_n$ repeat sites. According to karyotype and used species specific FISH markers we distinguished tadpoles of *P. lessonae* (h), *P. ridibundus* (b,g,k,n), diploid hybrids (a,f,i,j,l,m) and triploid hybrids with LLR (d,e) and RRL (c) genotypes. Crosses IDs correspond to Figs 1, 2 and S2 Table. Scale bars = 10 μm.
(TIF)

**S2 Fig.** Identification of tadpoles from crossings of triploid hybrid females with *P. lessonae* males (a, b), *P. ridibundus* males (d), diploid hybrid males (e-k) as well as crosses of hybrid males with *P. lessonae* (c, m) and *P. ridibundus* (l) females. Metaphase chromosomes of tadpoles after FISH with (TTAGGG)$_5$ (a-c,e-l) and RrS1 (d,m) probes. Arrows indicate interstitial (TTAGGG)$_n$ repeat sites. According to karyotype and used species specific FISH markers we distinguished tadpoles of *P. lessonae* (a,c), *P. ridibundus* (f,h,j), diploid hybrids (b,d,e,g,i,l,m) and triploid hybrids with LLR genotype (k). Crosses IDs correspond to Figs 1, 2 and S2 Table. Scale bars = 10 μm.
(TIF)

**S1 Table. List of studied European water frogs from the population systems of E type in Poland.**
(XLSX)

**S2 Table. List of crossed individuals and analyzed tadpoles with the indication of their genotypes.**
(XLSX)

## Acknowledgments

Authors would like to thank Vladislav Vasiullin for the help with the preparation of illustrations and Alena Kulikova for the help with animal collection. The authors acknowledge resource centers "Environmental Safety Observatory" and "Molecular and Cell Technologies" (Saint-Petersburg State University) for the access to experimental equipment.

## Author Contributions

**Conceptualization:** Dmitrij Dedukh, Alla Krasikova.

**Data curation:** Magdalena Chmielewska, Maria Ogielska, Alla Krasikova.

**Formal analysis:** Sergey Riumin, Krzysztof Kolenda.

**Funding acquisition:** Dmitrij Dedukh, Maria Ogielska.

**Investigation:** Dmitrij Dedukh, Sergey Riumin, Krzysztof Kolenda, Magdalena Chmielewska, Beata Rozenblut-Kościsty, Mikołaj Kaźmierczak.

**Methodology:** Dmitrij Dedukh, Sergey Riumin, Krzysztof Kolenda, Magdalena Chmielewska, Beata Rozenblut-Kościsty, Mikołaj Kaźmierczak, Maria Ogielska.

**Project administration:** Dmitrij Dedukh, Alla Krasikova.

**Resources:** Dmitrij Dedukh, Maria Ogielska.

**Supervision:** Dmitrij Dedukh, Alla Krasikova.

**Visualization:** Dmitrij Dedukh, Sergey Riumin.

**Writing – original draft:** Dmitrij Dedukh.

**Writing – review & editing:** Sergey Riumin, Krzysztof Kolenda, Magdalena Chmielewska, Beata Rozenblut-Kościsty, Maria Ogielska, Alla Krasikova.

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
