## [Decision Letter · Decision Letter 0]

28 Mar 2022

PONE-D-22-04633Maintenance of pure hybridogenetic water frog populations: genotypic variability in progeny of diploid and triploid parentsPLOS ONE

Dear Dr. Dedukh,

Thank you for submitting your manuscript to PLOS ONE. After careful consideration, we feel that it has merit but does not fully meet PLOS ONE’s publication criteria as it currently stands. Therefore, we invite you to submit a revised version of the manuscript that addresses the points raised during the review process.

Both reviewers were positive about your manuscript. I also can comment that the quality of the work, especially FISH results, is high. However, please address all the comments of both reviewers. ==============================

We look forward to receiving your revised manuscript.

Kind regards,

Igor V. Sharakhov

Academic Editor

PLOS ONE

Journal Requirements:

[Authors would like to thank Vladislav Vasiullin for the help with the preparation of illustrations and Alena Kulikova for the help with animal collection. The authors acknowledge resource centers “Environmental Safety Observatory” and “Molecular and Cell Technologies” (Saint-Petersburg State University) for the access to experimental equipment. The work of D.D. (grant holder) and S.R. was supported by grant from the Russian Science Foundation № 20-74-00030 (including expenses connected with cytogenetic analysis, microscopy, and scheme preparation). The work of M.O. (grant holder), M.C., B.R.-K., M.K. and K.K. was partially financed from the grant from Polish National Science Centre no.NCN 2012/07/B/NZ3/02563. The authors declare that the funding body had no role in the design of the study or the collection, analysis, and interpretation of the data or in writing the manuscript.]

 [D.D. Russian Science Foundation (https://rscf.ru/en/) № 20-74-00030

M.O. Polish National Science Centre (https://www.ncn.gov.pl/en) no.NCN 2012/07/B/NZ3/02563

The funders had no role in study design, data collection and analysis, decision to publish, or preparation of the manuscript.]

Reviewers' comments:

Reviewer's Responses to Questions

**Comments to the Author**

1. Is the manuscript technically sound, and do the data support the conclusions?

Reviewer #1: Yes

Reviewer #2: Yes

2. Has the statistical analysis been performed appropriately and rigorously? 

Reviewer #1: N/A

Reviewer #2: N/A

3. Have the authors made all data underlying the findings in their manuscript fully available?

Reviewer #1: Yes

Reviewer #2: Yes

4. Is the manuscript presented in an intelligible fashion and written in standard English?

Reviewer #1: Yes

Reviewer #2: Yes

5. Review Comments to the Author

Reviewer #1: Dear Dr. Krasikova,

I have carefully read your manuscript entitled "Maintenance of pure hybridogenetic water frog populations: genotypic variability in progeny of diploid and triploid parents". I believe this paper contains new important information on frog genotypes in hybridogenic populations from Poland, and therefore it can be published in PLOS ONE. However, I have a few suggestions which could apparently improve the manuscript. Specifically, material from the northwestern site of Wysoka Kamieńska is discussed in the main text, as opposed to Kamień Pomorski and Wysoka Kamieńska in the Supporting Information section. In addition, a particular verb, e.g. "obtained", is missing from the fourth line of the third paragraph on page 15 of the PDF file of the manuscript. Moreover, the volume and/or page numbers for the references no. 17 and 35 are missing. Please also check publication data for the reference no. 54 (apparently published in Annales Zoologici in Warsaw, Poland). Finally, the author's name for the reference no. 49 (a PhD thesis?) is given twice.

Reviewer #2: The hybridogenetic water frog populations analysed in this work provide an unique model for studying such a complex processes as polyploidisation, chromatin elimination and hybrid reproduction which are of great importance for evolution and speciation. The manuscript represents the results of scrupulous work on crossing and cytogenetic analysis of hybrid frogs from self-maintaining E system of diploid and triploid hybrids (655 tadpoles from 27 crosses). The cytogenetic method used in the study (karyotyping and FISH with species-specific markers) is sufficient and allows authors to identify genome composition of tadpoles. However, I have a couple minor suggestions for presenting and discussing the material.

Figures 1 and 2 clearly illustrate the results of experiments, however they largerly duplicate highly descriptive text in the Results section. Moreover, the summary of these results is nicely represented in Figure 4. I would probably recommend to move Figures 1 and 2 to the Supplementary material.

I believe that the manuscript would benefit if the authors put more emphasis on the unique results and significance of their work. Thus, authors reveal deviations from the "classical hybridogenetic reproduction program" as well as variable survival rate of particular progeny genotypes. What does this mean for understanding the maintenance of pure hybrid population systems? Do I understand correctly that one of the key findings of the work is that the genotype ratio in this E system is probably maintain due to postzygotic barriers (unviability of the certain progeny genotypes) rather than prezygotic (froming certain types of gametes)? I recommend this would be more clearly written.

I am not sure that the word "mechanism" is appropriate in this sentence of the Abstract: "To investigate the mechanism of diploid and triploid hybrid frog reproduction, we crossed these hybrid animals from two separate pure hybrid populations located in Poland." In this work, neither the mechanisms of elimination of one of the parental genomes, nor the mechanisms of choosing which of the genomes will be eliminated, nor the causes for the ratio of gametes of different types are addressed. However, brief discussion of these problems would help place the work in a broader context.

Unfortunately, I could not access Supporting Information (Table S1).

From the technical side, the manuscript lacks line and page numbering.

The manuscript meets PLOS ONE criteria for publication and can be published after minor revision.

6. PLOS authors have the option to publish the peer review history of their article (what does this mean?). If published, this will include your full peer review and any attached files.

Reviewer #1: No

Reviewer #2: No

---

## [Author Response · Author response to Decision Letter 0]

1 May 2022

Reviewer #1: I have carefully read your manuscript entitled "Maintenance of pure hybridogenetic water frog populations: genotypic variability in progeny of diploid and triploid parents". I believe this paper contains new important information on frog genotypes in hybridogenic populations from Poland, and therefore it can be published in PLOS ONE. 

Response: Thank you for saying nice words about the manuscript and appreciating our work. We have corrected the imprecisions throughout the text and figures according to your suggestions. 

However, I have a few suggestions which could apparently improve the manuscript. Specifically, material from the northwestern site of Wysoka Kamieńska is discussed in the main text, as opposed to Kamień Pomorski and Wysoka Kamieńska in the Supporting Information section.

Response: Thank you for showing this imprecision. We corrected the name of the locality in the Supporting Information section.

In addition, a particular verb, e.g. "obtained", is missing from the fourth line of the third paragraph on page 15 of the PDF file of the manuscript.

Response: We have added missing verb according to your suggestion.

Moreover, the volume and/or page numbers for the references no. 17 and 35 are missing. Response: Thank you for this point. We have added page numbers for mentioned references.

Please also check publication data for the reference no. 54 (apparently published in Annales Zoologici in Warsaw, Poland). 

Response: Thank you for the pointing this imprecision. We corrected it. 

Finally, the author's name for the reference no. 49 (a PhD thesis?) is given twice.

Response: We deleted additional author name.

Reviewer #2: The hybridogenetic water frog populations analysed in this work provide an unique model for studying such a complex processes as polyploidisation, chromatin elimination and hybrid reproduction which are of great importance for evolution and speciation. The manuscript represents the results of scrupulous work on crossing and cytogenetic analysis of hybrid frogs from self-maintaining E system of diploid and triploid hybrids (655 tadpoles from 27 crosses). The cytogenetic method used in the study (karyotyping and FISH with species-specific markers) is sufficient and allows authors to identify genome composition of tadpoles. However, I have a couple minor suggestions for presenting and discussing the material.

Response: Thank you very much for your work and appreciating our results. We have changed manuscript to address your suggestions in most cases and we provided explanations bellow in detailed responses.

Figures 1 and 2 clearly illustrate the results of experiments, however they largerly duplicate highly descriptive text in the Results section. Moreover, the summary of these results is nicely represented in Figure 4. I would probably recommend to move Figures 1 and 2 to the Supplementary material.

Response: You raise an important point. Thank you for understanding that Figures 1 and Figure 2 are important to demonstrates different examples of all types of tadpoles obtained from various crosses for two analysed hybrid systems. However, we decided to keep both Figures in the main text of the manuscript as it will simplify reading the text and comparing results for each of studied populations. In contrast, Figure 4 gives just the summary of those results and missing the information about multiple crosses of some individuals.

I believe that the manuscript would benefit if the authors put more emphasis on the unique results and significance of their work. Thus, authors reveal deviations from the "classical hybridogenetic reproduction program" as well as variable survival rate of particular progeny genotypes. What does this mean for understanding the maintenance of pure hybrid population systems? Do I understand correctly that one of the key findings of the work is that the genotype ratio in this E system is probably maintain due to postzygotic barriers (unviability of the certain progeny genotypes) rather than prezygotic (froming certain types of gametes)? I recommend this would be more clearly written.

Response: Thank you for raising these important questions. We clarified our conclusions according to your comment and added the information in the abstract, discussion and conclusions.

In Abstract:

Lines 32-34. “In addition, we found a variable survival rate of particular progeny genotypes when we crossed hybrid females with different parents suggesting the important role of postzygotic barriers on the maintenance of pure hybrid systems.”

In Discussion section: 

Lines 387-389. “Thus, the viability of the certain progeny genotypes indicates the strong effect of postzygotic rather than prezygotic barriers on the maintenance of studied hybrid systems.”

In Conclusions: 

Lines 424-429. “Moreover, we observed sex-specific differences in the contribution to the progeny in both diploid and triploid hybrids. In addition, we detected preferential variability in the survival rate of particular genotypes, depending on which species the progeny is crossed with, suggesting that postzygotic barriers play an important role on the maintenance of hybrid systems.”

I am not sure that the word "mechanism" is appropriate in this sentence of the Abstract: "To investigate the mechanism of diploid and triploid hybrid frog reproduction, we crossed these hybrid animals from two separate pure hybrid populations located in Poland." In this work, neither the mechanisms of elimination of one of the parental genomes, nor the mechanisms of choosing which of the genomes will be eliminated, nor the causes for the ratio of gametes of different types are addressed. However, brief discussion of these problems would help place the work in a broader context.

Response: According to your suggestions, we corrected the abstract and included additional paragraph where we discuss hypothetical mechanisms concerning elimination of parental genomes in studied systems. 

In Abstract:

Lines 19-22. “To investigate how diploid and triploid hybrid frog reproduce in populations without parental species, we crossed these hybrid animals from two separate pure hybrid populations located in Poland.”

In Discussion: 

Lines 390-403. “Comparative analysis of crosses results and hybrids gametogenesis in E [28, 29, 59], R-E [36–38, 50, 51, 60, 61], L-E [17, 39, 42, 54, 56, 62, 63] and R-L-E [42] systems, and direct observations of eliminated genome in hybrid tadpoles [42, 43] clearly show that P. lessonae genome is preferentially eliminated in contrast to P. ridibundus genome. Nevertheless, striking variability of gametes produced by diploid and triploid hybrids in E systems suggest aberrancies in the genome elimination and endoreplication during hybrids frog gametogenesis (this study, [28, 29, 59]). Such aberrancies may be caused by the genetic introgression from other water frog species into the genetic background of P. ridibundus and thus completely or partially prevent hybridogenetic reproduction [63]. Additionally, the ability of P. lessonae genome to persist elimination can cause competition between different genomes to be eliminated or retained, resulting in gamete variability in one individual. Interestingly, that R-E systems in the central Europe, in which diploid hybrid males produce two types of sperm, have a single origin suggesting the possibility of latter scenario [50, 52]. Nevertheless, the mechanism of genome elimination in water frogs requires further investigation.”

Unfortunately, I could not access Supporting Information (Table S1).

Response: We apologize for this inconvenience and uploaded all files in the revised version. 

From the technical side, the manuscript lacks line and page numbering.

Response: We apologize for this inconvenience. We corrected it in the revised version.

---

## [Editor Report · Decision Letter 1]

3 May 2022

Maintenance of pure hybridogenetic water frog populations: genotypic variability in progeny of diploid and triploid parents

PONE-D-22-04633R1

Dear Dr. Dedukh,

We’re pleased to inform you that your manuscript has been judged scientifically suitable for publication and will be formally accepted for publication once it meets all outstanding technical requirements.

Kind regards,

Igor V. Sharakhov

Academic Editor

PLOS ONE
---

## [Editor Report · Acceptance letter]

14 Jun 2022

PONE-D-22-04633R1 

Maintenance of pure hybridogenetic water frog populations: genotypic variability in progeny of diploid and triploid parents 

Dear Dr. Dedukh:

I'm pleased to inform you that your manuscript has been deemed suitable for publication in PLOS ONE. Congratulations! Your manuscript is now with our production department. 

Kind regards, 

on behalf of

Dr Igor V. Sharakhov 

Academic Editor

PLOS ONE